# Therapeutic Effect of C-Vx Substance in K18-hACE2 Transgenic Mice Infected with SARS-CoV-2

**DOI:** 10.3390/ijms241511957

**Published:** 2023-07-26

**Authors:** Hivda Ulbegi Polat, Irem Abaci, Arzu Tas Ekiz, Ozge Aksoy, Fatma Betul Oktelik, Vuslat Yilmaz, Saban Tekin, Alper Okyar, Oral Oncul, Gunnur Deniz

**Affiliations:** 1TUBITAK Marmara Research Center, Kocaeli 41470, Türkiye; hivda.polat@tubitak.gov.tr (H.U.P.); irem.abaci@tubitak.gov.tr (I.A.); arzu.tas@tubitak.gov.tr (A.T.E.); ozgeaksoy.356@gmail.com (O.A.); 2Department of Biotechnology, Institute of Biotechnology, Gebze Technical University, Kocaeli 41400, Türkiye; 3Department of Molecular Biology and Genetics, Institute of Sciences, Yildiz Technical University, Istanbul 34220, Türkiye; 4Department of Immunology, Aziz Sancar Institute of Experimental Medicine, Istanbul University, Istanbul 34452, Türkiye; oktelikbetul@gmail.com; 5Department of Neuroscience, Aziz Sancar Institute of Experimental Medicine, Istanbul University, Istanbul 34452, Türkiye; vuslat.yilmaz@istanbul.edu.tr; 6Department of Medical Biology, University of Health Sciences, Istanbul 34865, Türkiye; saban.tekin@sbu.edu.tr; 7Department of Pharmacology, Faculty of Pharmacy, Istanbul University, Istanbul 34452, Türkiye; aokyar@istanbul.edu.tr; 8Department of Infectious Diseases and Clinical Microbiology, Istanbul Faculty of Medicine, Istanbul University, Istanbul 34452, Türkiye; oraloncul@istanbul.edu.tr

**Keywords:** C-Vx, SARS-CoV-2, COVID-19, antiviral, cytokines, neutralization

## Abstract

C-Vx is a bioprotective product designed to boost the immune system. This study aimed to determine the antiviral activity of the C-Vx substance against SARS-CoV-2 infection. The effect of C-Vx in K18-hACE2 transgenic mice against the SARS-CoV-2 virus was investigated. For this purpose, ten mice were separated into experimental and control groups. Animals were infected with SARS-CoV-2 prior to the administration of the product to determine whether the product has a therapeutic effect similar to that demonstrated in previous human studies, at a histopathological and molecular level. C-Vx-treated mice survived the challenge, whereas the control mice became ill and/or died. The cytokine-chemokine panel with blood samples taken during the critical days of the disease revealed detailed immune responses. Our findings showed that C-Vx presented 90% protection against the SARS-CoV-2 virus-infected mice. The challenge results and cytokine responses of K18-hACE2 transgenic mice matched previous scientific studies, demonstrating the C-Vx’s antiviral efficiency.

## 1. Introduction

Coronavirus Disease 2019 (COVID-19) is a severe acute respiratory syndrome caused by SARS-CoV-2 [1]. In December 2019, cases of viral pneumonia of an unknown cause emerged in Wuhan, China’s Hubai province. It spread rapidly around the world and in March 2020, the World Health Organization (WHO) declared that the COVID-19 disease was a global pandemic [2,3]. The virus showed 79% and 50% homology to the genomes of SARS-CoV, which emerged in 2002, and MERS-CoV, which appeared in 2012, respectively, and therefore the new coronavirus was named SARS-CoV-2 [4].

The clinical course of the disease varies from asymptomatic infection to multi-organ failure and death. According to a clinical study, the most common symptoms of the disease were cough and fatigue (56.6%), taste disturbance (35.7%), myalgia (34.3%) and fever (33.6%). When the two groups with positive and negative test results were compared, taste disorders, olfactory disorders and diarrhea were found to be significantly higher in the COVID-19-positive group [5,6]. The viral infection path starts with the binding of the SARS-CoV-2 spike protein to human angiotensin I-converting enzyme 2 (hACE2) and the targeting of the virus to type II pneumocytes in the lung. This initiate pathological lesions in the lung. It then progresses with inflammation in the target organ and respiratory distress [7,8]. Severe and critical symptoms include multiple organ dysfunction, respiratory failure, chronic renal failure, heart complications, stroke, Guillain-Barre syndrome, and others [8,9]. Most patients recover from the acute phase of the disease; however, some patients may show some symptoms months after recovery, and this condition is called “Long COVID”. Long COVID is characterized by long-term complications or ongoing symptoms after the typical COVID-19 recovery period; almost every organ system can be affected by this disease, but the most common effects are seen in the respiratory, nervous and cardiovascular systems [10]. According to WHO, post-COVID-19 condition occurs in persons with a history of probable or confirmed SARS CoV-2 infection, usually occurring 3 months after the onset of COVID-19 with symptoms persisting for at least 2 months, which cannot be explained by an alternative diagnosis [10,11]. Symptoms include fatigue, weakness, musculoskeletal pain, headache, shortness of breath, parosmia, anosmia, low fever, anemia, and cognitive dysfunction. Some patients experience multisystem inflammatory syndrome in which severe damage to organs is observed [8].

In early studies, it was thought that different antiviral vaccines have an effect on COVID-19 disease. Traditionally, vaccinations are intended to train the adaptive immune system by generating an antigen-specific immune response. It is also suggested that certain vaccines provide protection against other infections through trained immunity for 1 to 5 years. For example, some hypotheses and statistical studies have been made that measles, mumps and rubella vaccines can prevent or reduce the severity of COVID-19 [12,13]. Despite a large number of ongoing scientific and clinical studies, drugs that specifically target SARS-CoV-2 have not yet been identified or approved for clinical use. Although such a non-specific treatment protocol is applied to patients with SARS-CoV-2 infection to alleviate symptoms and improve prognosis, it has been observed that controlling the progression of the infection mostly depends on the patient’s own immune defense [14].

As we now know better, in the SARS-CoV-2 infection, the innate and adaptive immune systems are both activated. Natural killer (NK) and innate lymphoid cells (ILC) form part of the innate immune system, where they act as a first-line defense; CD4^+^ and CD8^+^ T cells, which are adaptive immune system cells, play important roles in antiviral responses [15,16,17]. The innate immune response is able to recognize viral patterns (Pathogen-Associated Molecular Patterns—PAMPs) through its receptors (Toll-Like Receptors—TLRs). As a result of receptor interaction, various transcription factors such as the intracellular Nuclear Factor Kappa-light-chain-enhancer of activated B cells (NFκB) are activated and cause the expression of pro-inflammatory factors. In COVID-19 disease, increased levels of pro- and anti-inflammatory cytokines and chemokines, which induce a high degree of immune response that causes cytokine release syndrome (CRS), were reported [18]. Many studies showed that there were defects also in the functions of innate and adaptive immune cells and reduced cell numbers in patients infected with COVID-19. On the other hand, in SARS-CoV-2 infection, humoral immune responses develop as well as cellular responses. B cells produce specific antibodies against the antigenic determinants of the virus. Studies show that SARS-CoV-2-specific IgG antibody responses develop after infection or vaccination. Considering all these immunological parameters, regulatory treatments or vaccines are needed after infection with SARS-CoV-2.

The C-Vx was created by employing a measles viral vector to stimulate anti-tumor immune responses [19]. The drug’s components were updated during the SARS-CoV-2 outbreak to include the recombinant SARS-CoV-2 gene fragment, adjuvant, vitamins, and attenuated mumps and rubella viruses and developed for use in the treatment of COVID-19. This study aimed to assess the potential contribution of mumps-rubella vaccinations to the prevention of COVID-19 disease, as described above, by utilizing the projected effect on SARS-CoV-2. To determine the effect of C-Vx in mice, we conducted in vivo dose–response and toxicity analysis of C-Vx in mice to see if it helps to reduce the ability of immune cells to function and cytokine storm.

## 2. Results

### 2.1. Challenge Experiment for the Determination of the Doses

In the first challenge experiment, in Group 1 (100 µL/mouse, n = 4), three out of four mice died of the disease and one survived. Between 14–23% of body weight loss occurred in dead mice. In Group 2 (130 µL/mouse, n = 4), three mice showed disease symptoms, and two of them died before completing the experiment. The two mice that died had a body weight loss of 25% and 29%, respectively. The animals in Group 3 (180 µL/mouse, n = 4) did not show any clinical symptoms. Only two mice lost 3–5% of body weight. However, in real-time PCR experiments, a low Ct (<20), which indicates high viral load, was detected in three out of four mice. Two mice died at the end of the experiment in the control group (n = 4). PCR Ct in the N1 and N2 genes was inconsistent in the remaining two mice. However, one of these mice displayed symptoms related to illness, and both mice lost 14% to 29% of their body weight.

Because the 180 µL/mouse dose produced a better response in the first study, it was repeated in the second study by increasing the number of mice. Furthermore, 220 µL/mouse and 250 µL/mouse doses were used. In the second challenge experiment, in Group 1 (180 µL/mouse, n = 6), four out of six mice, of which there would be six mice in each group, showed clinical symptoms, with a body weight loss of 9–19%. The remaining two mice had no clinical symptoms and overcame the disease. High-positive viral load was detected in RT-PCR in four infected mice, while viral load was not detected in two healthy mice. Similarly, two out of six mice in Group 2 (220 µL/mouse, n = 6) showed severe clinical symptoms, with 22% and 34% body weight loss. High viral load was detected in these two mice, while the remaining four mice overcame the disease, and no viral load was detected. In Group 3 (250 µL/mouse, n = 6), one out of six mice showed clinical symptoms, with approximately 28% body weight loss. High viral load was detected in this mouse, while the remaining five mice had no clinical symptoms, no significant weight loss, and no viral load was detected. In Group 3 (250 µL/mouse) that received the C-Vx against SARS-CoV-2 virus, the animals overcame the disease. One out of six mice in the control group that did not receive C-Vx showed resistance and defeated the disease. The remaining five mice developed clinical symptoms, with a body weight loss of 14–22%. At the same time, a higher viral load was detected in these five mice (Ct < 20). A high viral load was detected in all mice when analyzed by RT-PCR. The Ct values are presented in Appendix A.

In the last part of study, since the mice given a dose of 250 µL/mouse recovered from the disease and most of the mice did not show clinical symptoms, the study was continued with a dose of 250 µL/mouse and control with 10 mice per group. At the end of the experiment, two of the ten mice that were given C-Vx had lost 16% and 34% of their body weight, respectively. On the sixth day of the experiment, one of these animals was found as it was about to die from infection, and pathology was performed. This animal displayed clinical symptoms such as weight loss, hunched posture, and burring in the eyes. On the other hand, the other animal showed no clinical symptoms other than weight loss. The remaining animals (80%) completed the experiment in good health, and their weight increased until the experiment’s conclusion.

Weight loss was seen in control animals four days after infection. At 7 dpi (days post injection), one of the control animals was discovered dead in the cage. The remaining nine animals (80%) had dramatic weight loss (14–38%) and clinical symptoms at 8 dpi. Control animals were exterminated at 8 dpi and autopsied for animal welfare reasons (Figure 1A). The lungs were evaluated in gross pathology in terms of color, tissue integrity, organ appearance, and size. The lungs of the control group animals had a different size of pneumonia, whereas the lungs of the C-Vx experimental-group mice were clear and showed no signs of infection (Figure 1B). The gross pathological inflammation score of the lungs from the control and C-Vx groups is shown in Figure 1C.

In the treated group, histological examination of lung tissues revealed the healing and restoration of normal tissue components. Lung sections were analyzed with hematoxylin and eosin, and the inflammation of the lung was semi-quantitatively scored between 0 and 3 (0: no inflammation, 1: low, 2: medium, 3: high). In the control group’s lungs interstitial inflammatory-cell infiltration, alveolar septal thickening and more parenchymal infiltration at the peribronchiolar regions were observed compared to C-Vx-treated animals, which had relatively healthy lungs (Appendix A). The histopathology inflammation score of the lungs from the control and C-Vx groups is shown in Figure 1D. The viral loads in the harvested lungs were determined using RT-PCR, with two primers that targeted the SARS-CoV-2 virus nucleocapsid gene (Appendix A) [20,21,22]. According to PCR Ct values, eight out of the ten animals in the control group had a high viral load (Ct values around 20). One of the remaining animals died due to the disease, but one survived with individual variation, indicating a low viral load (Ct = 37). In contrast to the control group, nine out of ten C-Vx-treated animals did not become sick, and no viral load was detected (Ct = 40–42). The remaining one animal, however, had a viral load (Ct = 16).

### 2.2. Increased Cytokine Levels in K18-hACE2 Mice Sera Treated with the C-Vx Substance

Cytokines are proteins that are important in controlling the growth and activity of immune cells and grouped as pro-inflammatory and anti-inflammatory cytokines. In this study, the levels of 18 cytokines and chemokines were investigated in mice serum using multiple measurement methods. Although there were no differences in the cytokine or chemokine levels at baseline and on the 3rd day (Figure 2A,B), it was determined that the levels of various cytokines and chemokines were increased significantly in the C-Vx-treated group compared to the control group on the 15th day (last day) (Figure 3B). Among these cytokines, IFN-γ has a role in stimulating NK cells, neutrophils and especially macrophages. IFN-γ is classically considered as a pro-inflammatory cytokine group. IFN-γ, IL-1β, IL-5, IL-17α, which are pro-inflammatory cytokines, significantly increased in the C-Vx-treated group compared to the control group (*p* = 0.0025, *p* = 0.0288, *p* = 0.0001 and *p* = 0.0001, respectively) (Figure 3B). In addition, IL-4 and IL-13 anti-inflammatory cytokines were found to be increased (*p* = 0.0344 and *p* = 0.0045, respectively) in the C-Vx-treated group (Figure 3B). However, only increased GM-CSF and decreased MCP1 were observed in the C-Vx-treated group on day 5 (Figure 3A).

In the current study, besides cytokines, the chemokine levels of mice LIX, neutrophil and monocyte chemotactic proteins in the CXC family of cytokines (CXCL5), CXCL1/KC and MCP-1/CCL2 chemokines were also evaluated. MCP-1 is one of the important chemokines that regulates the migration of monocytes/macrophages. LIX, KC and MCP-1 chemokines showed similar responses to the cytokine profile. On the 15th day, compared to the control group, C-Vx-treated groups were found to have a significantly higher chemokine response (*p* = 0.0020, *p* = 0.0002 and *p* = 0.0003, respectively) (Figure 3B). Overall, our results show that many cytokines and chemokines are induced and show patterns of upregulation in the inflammatory response to SARS-CoV-2 in K18-hACE2 mice.

## 3. Discussion

Several studies have been conducted in the literature which model K18-ACE2 transgenic mice against COVID-19 disease. Winkler et al. observed that after the intranasal infections of K18-ACE2 mice at 4 dpi and 7 dpi, the animals lost serious weight and had high viral loads at 2 dpi, 4 dpi, and 7 dpi [23]. According to the work of Bao et al., the highest amount of virus was detected in the lungs of hACE2 mice at 3 dpi; then, the amount of virus decreased at 5 dpi and continued to decrease at 7 dpi [24]. According to the studies, if K18-ACE2 transgenic mice are exposed to the virus every other day, the viral load begins to decrease after the 5th day, and after the 7th to 9th day, the viral load drops to a non-significant level. The virus was discovered to have vanished in a matter of days. As a result, in this study, the intranasal administration of B1.1.7 virus with a good TCID_50_ value was performed on consecutive days to determine C-Vx protection and to establish a complete disease in our control animals [20,21,25]. Blood was drawn before the start of the infection, in addition to at 3 dpi and 5 dpi, which are critical days of the infection, to demonstrate the cytokine and chemokine responses of C-Vx in the model animal. Concurrently, cardiac blood was drawn prior to the autopsy to demonstrate the improvement in these model animals. The challenge experiment in our study was designed similarly to that in humans due to the therapeutic aspect of C-Vx. Even though C-Vx was administered parenterally two days after infection, it was determined that it provided excellent protection. Vaccination protects the body from various diseases by increasing natural immunity, according to studies conducted during the COVID-19 process. According to some reports, innate immunity provided by some considered-safe childhood vaccines, such as Bacillus Calmette-Guerin (BCG), Polio, Hemophilus influenza type-B, Measles-Mumps and Rubella (MMR), and pneumococcus, can provide significant protection against COVID-19 [12,26,27]. One of the reasons for the low prevalence of COVID-19 in children appears to be the high levels of immunity provided by childhood vaccinations [14]. However, the use of the relevant childhood vaccines, particularly in conjunction with the MMR vaccine, indicates that the MMR vaccine provides most of the protection against COVID-19 disease. There are other hypotheses that the SARS-CoV-2 surface protein shares 29% of its sequence with the rubella virus [27,28]. Therefore, there are theories that the MMR vaccine’s rubella component offers particular protection against COVID-19. The relationship between the MMR vaccine and COVID-19 has yet to be established and is currently considered theoretical. It has been reported, however, that in countries where MMR vaccination rates are high, the prevalence of COVID-19 disease and the mortality rate are lower [14]. Because COVID-19 is similar to other respiratory diseases such as measles, rubella, and mumps, immunity following MMR vaccination is thought to be one of the reasons why children do not contract COVID-19 [14]. According to the literature, effectiveness of C-Vx in the challenge experiments is due to the MMR vaccine it contains [12,26,27]. The C-Vx group did not lose weight as compared to the control group, although their lungs were microdamaged, which was confirmed by histological findings [20,22,25,29]. C-Vx was found to be effective in K18-ACE2 transgenic mice infected with SARS-CoV-2. The mechanism of action of the C-Vx product will be thoroughly studied in the next step of our research.

## 4. Material and Methods

The Jackson Laboratory in the United States provided the K18-hACE2 [B6.Cg-Tg (K18-hACE2) 2Prlmn/J] transgenic mice were used in this study. TUBITAK MRC, Life Sciences, Medical Biotechnology Unit in Gebze, Kocaeli, Turkiye, approved 8–10-week-old female K18-hACE2 transgenic mice (20–22 g body weight). All procedures in this study involving animals were reviewed and approved by the Institutional Biosafety Committee and Institutional Animal Care and Use Committee (HADYEK-16563500-111-1807); all the experiments were conducted in compliance with all relevant ethical regulations. The experiments were conducted in Biosafety Level 3 (BSL3) and animal BSL3 (ABSL3) facilities at TUBITAK MRC Life Sciences.

The C-Vx product used in this study was developed by Hamida Pharma, Lake Forest, CA, USA together with Miracle Labs, Istanbul, Turkey. With the emergence of the COVID-19 epidemic, the scientific team made changes to the formula of the substance. Currently C-Vx is patented as an active immunostimulant [19].

### 4.1. Determination of C-Vx Dosage

The effectiveness of the viral dose on mice was determined using the challenge experiment and LD50 criteria. When more than 50% survival was achieved in the intraperitoneal doses administered in the challenge test groups, a higher dose was administered. The volume-per-weight calculation was not performed because the newly developed C-Vx is a biological product [16]. As a result, in each group, the same volume of C-Vx was given to animals of the same weight living in the same environments [20,30,31].

For the challenge experiments, three different doses of C-Vx (100 µL/mouse, 130 µL/mouse, 180 µL/mouse, n = 4 for each group) were applied to mice, and to the control group (n = 4) only PBS was given. Because survival was good at the 180 µL/mouse dose, trials at the 180 µL/mouse dose were conducted again, and two different doses were conducted concurrently. In the second study, doses of 180 µL/mouse, 220 µL/mouse, and 250 µL/mouse (n = 10) were administered. The experiment was carried out twice. In the last study, a dose of 250 µL/mouse was used because it provided the best response.

### 4.2. Intranasal Delivery for SARS-CoV-2 Infection in Mouse Models

The Ministry of Health Public Health Directorate supplied the SARS-CoV-2 virus strain B.1.1.7 alpha. The purpose of the study was to be therapeutic rather than preventive. The B.1.1.7 strain (alpha) of SARS-CoV-2 virus with a TCID_50_ value of 10^5^ was used in this study. Under anesthesia, the SARS-CoV-2 virus was administered intranasally for three days [20,31]. The treatment group received 250 µL of C-Vx intraperitoneally four hours after virus administration on the third day of infection and continued for four days (Figure 4). Similarly, animals in the placebo group received 250 µL of PBS. The experiment lasted a total of 15 days. After being infected with the virus, mice were monitored on a daily basis for morbidity (body weight) and mortality. Mice that lost more than 25% of their baseline body weight were considered to have reached the experimental endpoint and were exterminated.

Before the challenge experiment, blood samples were taken from the mice at different times for the cytokine and chemokine studies. Tail blood was sampled before viral inoculation at the beginning of the study from the experimental and control groups. An insulin injector was used to collect approximately 50 µL of blood from the lateral tail vein. In the period 24 h after the second viral inoculation, tail blood was drawn again. Since it is known that changes in cytokine and chemokine levels are observed at a significant level after 4–6 days of the disease, for control animals, this time interval is important [32]. Blood (0.5 mL) was taken from the animal while they were under anesthesia by inserting an insulin injector into the heart at a 20–30 degree angle just to the left of the xiphosternal midline. Mice were sacrificed by cervical dislocation following anesthesia with sevoflurane inhalation, and autopsies were conducted. Blood serum was obtained and stored at −20 °C for cytokine measurements, which required 25 µL of serum per well. All abdominal organs and thoracic cavity organs were examined with the naked eye during gross pathology. One half of each mouse lung was used for real-time PCR to compare viral load, and the other half was used for histopathological analysis.

Tissue samples were prepared for histopathological analysis by being preserved in 10% buffered formalin, dried in a succession of increasingly concentrated alcohol solutions, and embedded in paraffin wax. Sections were cut to a 5 µm thickness. Before staining, sections were deparaffinized with xylene and rehydrated in a series of alcohol solutions of decreasing concentration. Sections were stained with hematoxylin for 5 min and then rinsed in running tap water. After washing, they were stained with eosin for 5 min and rinsed in running tap water. The stained sections were dehydrated, cleared and mounted. The slides were observed under the Zeiss Axio Vert A1 microscope (Zeiss, Oberkochen, Germany) with Zen software 2.6 pro version (Zeiss, Oberkochen, Germany).

### 4.3. Isolation of Viruses and RNA Extraction for RT-PCR

After the mice’s lungs were harvested, and half of the lungs were separated for real-time analysis. Lung tissues were homogenized separately in 2 mL of PBS using an ultrasonic homogenizer at 70% amplitude for 90 sec (Bandelin HD2200.2, Berlin, Germany) for tissue homogenization. Tissue homogenates were centrifuged at 21.500× *g* for 10 min and supernatants were collected into 15 mL falcon tubes. Viral RNA was extracted with the QIAamp Viral RNA Mini kit (Qiagen, Germantown, MD, USA) according to the instructions of the manufacturer. The viral RNA detection was performed using SARS-CoV-2 nucleocapsid specific primer and probes detailed below with One Step PrimeScript III RT-PCR Kit (Takara, Kusatsu, Japan). All reactions were performed on a CFX96 Touch instrument (Bio-Rad, Hercules, CA, USA) with the following real-time PCR conditions: 52°C for 5 min, 95°C for 10 s, then 44 cycles of 95°C for 5 s and 55°C for 30 s. The primer and probe sequences that were used for RT-PCR are CDC recommended and FDA EUA approved NC1 and NC2 sequences. The target region was the Nucleocapsid (NC) gene of SARS-CoV-2. Primer and probe sequences were N1 Forward: 5′-GAC CCC AAA ATC AGC GAA AT-3′, N1 Reverse: 5′-TCT GGT TAC TGC CAG TTG AAT CTG-3′ N1 Probe: 5′-FAM-ACC CCG CAT TAC GTT TGG TGG ACC-BHQ1-3 N2 Forward: 5′-TTA CAA ACA TTG GCC GCA AA-3′ N2 Reverse: 5′-GCG CGA CAT TCC GAA GAA-3′ N2 Probe: 5′-FAM-ACA ATT TGC CCC CAG CGC TTC AG-BHQ1-3 [33].

### 4.4. Cytokine Measurement with LUMINEX

Blood samples were taken from the mice at day 0 (baseline), and the 3rd, 5th and 15th days in the treated-with-C-Vx and untreated groups. After the blood centrifugation, separated serum samples were kept at −80°C until use. The cytokine levels in serum samples were detected by using a MILLIPLEX® Mouse High Sensitivity T Cell Magnetic Bead Panel (Sigma Aldrich, St. Louis, MO, USA) kit. This 18 plex, consisting of Granulocyte macrophage colony-stimulating factor (GM-CSF), interferon gamma (IFN-γ), interleukin (IL)-1α, IL-1β, IL-2, IL-4, IL-5, IL-6, IL-7, IL-10, IL-12 (p70), IL-13, IL-17α, chemokine (C-X-C motif) ligand 1 (KC/CXCL1), lipopolysaccharide-induced CXC chemokine (LIX), monocyte chemoattractant protein-1 (MCP-1/CCL2), macrophage inflammatory protein 2 (MIP-2/CXCL2) and tumor necrosis factor alpha (TNF-α), was applied according to the manufacturer’s protocol. Minimum detectable concentration (MinDC) was calculated using MILLIPLEX® Analyst 5.1. Assay sensitivities (minimum detectable concentrations, pg/mL) per bead/analyte and the Luminex® Magnetic Bead Region are given in Table 1. The plate was run on a Luminex® 200™ 3.1 HTS instrument with XPOTENT software for LX100/LX200 (Luminex Corporate, Austin, Houston, USA).

### 4.5. Statistical Analysis

The normality of data distribution was determined by a Kolmogorov–Smirnov test. Student’s *t*-test or the Mann–Whitney U test was used for the pairwise comparisons of groups with normal and non-normal distribution, respectively. The data were presented as mean ± standard deviation. A two-sided *p* value < 0.05 was considered statistically significant. Statistical analyzes and graphs were performed with SPSS IBM 21.0 and GraphPad Prism 5 programs.

## Figures and Tables

**Figure 1 ijms-24-11957-f001:**
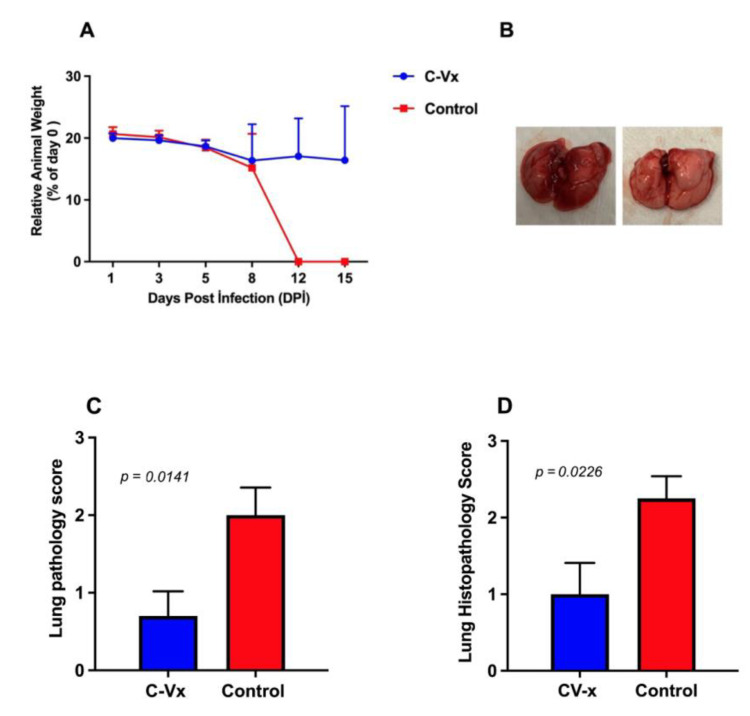
Neutralization effect of C-Vx drug in vivo. (**A**) Average change in body weight by time in control and C-Vx treated mice. Only 12th and 15th days were significant (*p* < 0.0001 for both days) between the groups. (**B**) Representative lung images dissected from control and C-Vx treated mice. (**C**) Gross pathology inflammation scores from control and C-Vx-treated mice lungs. Lung inflammation was scored from 0 to 3 (0: no inflammation, 1: low, 2: medium, 3: high). Results for the gross pathology in the lung were significant (*p* = 0.0141). (**D**) Histopathology inflammation score graph of the lungs from the control and C-Vx group. Results for the inflammation score in the lung were significant (*p* = 0.0226).

**Figure 2 ijms-24-11957-f002:**
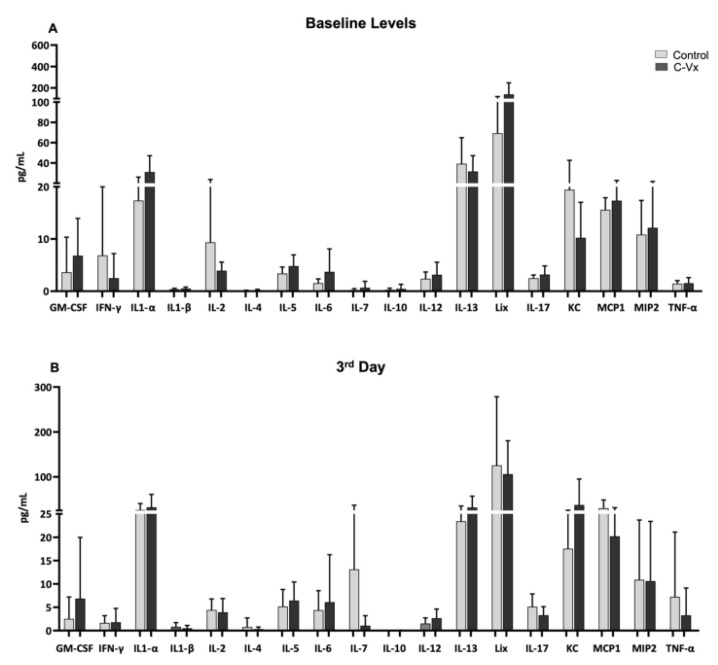
Determination of 18 different cytokines in mice serum samples. (**A**) Cytokine values were determined in serum samples taken from mice on the first day (baseline levels). The *p* value on the first day was not significant when the control and C-Vx groups were compared. (**B**) Cytokine values were determined in serum samples taken from mice on the 3rd day. A Mann–Whitney U test was used for statistical analysis in pairwise group comparisons. A *p* value < 0.05 was considered statistically significant.

**Figure 3 ijms-24-11957-f003:**
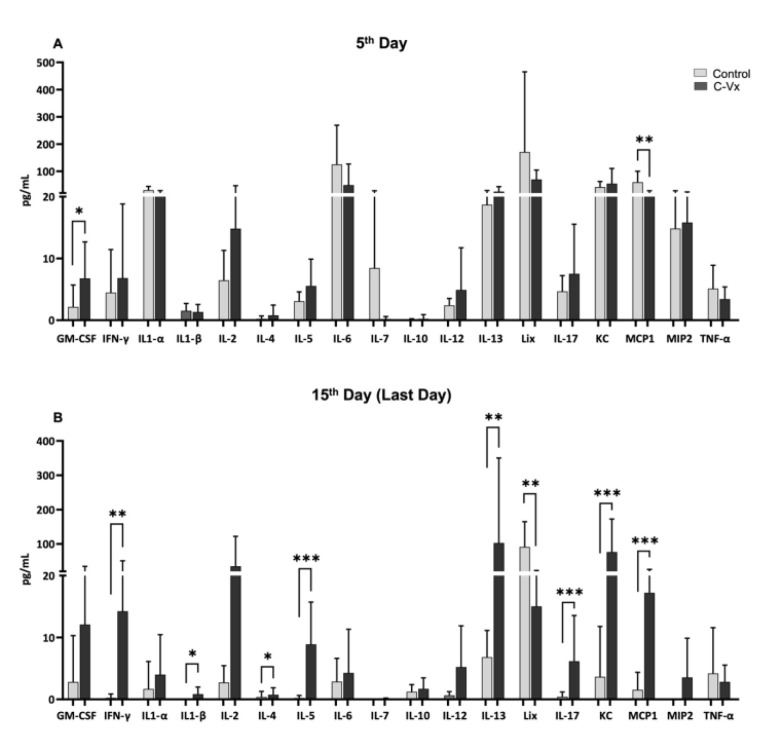
Determination of 18 different cytokines in mice serum samples. (**A**) Cytokine values were determined in serum samples taken from mice on the 5th day. On the 5th day, GMCSF cytokine levels were significantly higher in the C-Vx treated group compared to the control group. (**B**) Cytokine values were determined in serum samples taken from mice on the 15th day. It was observed that pro-inflammatory cytokines in the serum samples of the following days (15th day) were significantly higher in the C-Vx-treated group compared to the control group. A Mann–Whitney U test was used for statistical analysis in pairwise group comparisons. A *p* value < 0.05 was considered statistically significant. * *p* < 0.05, ** *p* < 0.01, *** *p* < 0.001.

**Figure 4 ijms-24-11957-f004:**
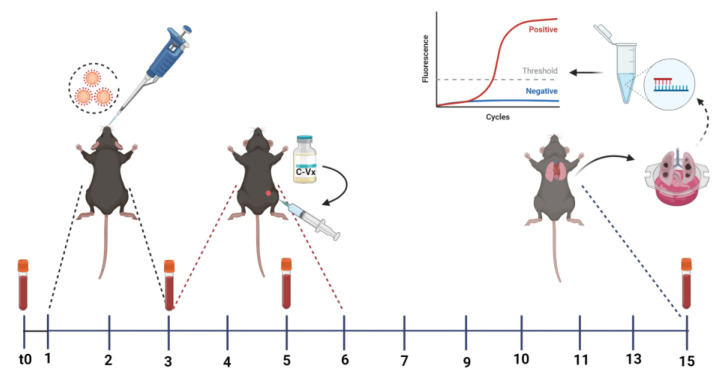
Schematic representation of in vivo experiment to test the efficacy of C-Vx in vivo. SARS-CoV-2 (Alpha) was delivered to mice that express human ACE2 (hACE2) for 3 days, followed by 4 days of C-Vx treatment beginning on day 3 of infection (n = 10/group). Created with BioRender.com.

**Table 1 ijms-24-11957-t001:** Assay sensitivities (minimum detectable concentrations, pg/mL) per bead/analyte and Magnetic Bead Region.

Bead/Analyte Name	MinDC (pg/mL)	MinDC+2SD (pg/mL)	Magnetic Bead Region
GM-CSF	5.33	7.27	15
IFN-γ	0.15	0.26	19
IL-1α	1.54	3.12	21
IL-1β	2.58	4.68	25
IL-2	0.80	1.34	26
IL-4	0.06	0.12	28
IL-5	0.53	1.00	30
IL-6	0.54	1.26	34
IL-7	0.98	2.48	36
IL-10	0.53	1.10	43
IL-12	1.29	2.82	47
IL-13	3.76	6.16	52
LIX	2.70	6.26	53
IL-17A	0.18	0.38	56
KC/CXCL1	0.45	0.91	61
MCP-1	3.00	6.20	62
MIP-2	9.06	11.14	73
TNF-α	0.41	0.76	77

## Data Availability

Data sharing is not applicable to this article.

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
