# Peer review of "Therapeutic Effect of C-Vx Substance in K18-hACE2 Transgenic Mice Infected with SARS-CoV-2"

_ijms, 2023, doi:10.3390/ijms241511957_

Round 1

Reviewer 1 Report

Thanks to the authors for providing this manuscript. This research demonstrates the therapeutic potential of C-Vx against coronavirus infection in a mouse model. The phenotypic data from the in vivo experiment in this study provides a valuable reference for readers.

Major points:

1.      The authors mentioned “it was determined that the levels of various cytokines and chemokines were increased significantly in the C-Vx treated group compared to the control group in the 15th day (last day) (Figure 5B).” at line 176-178. However, the results shown in Figure 5 appear to be opposite. The detected cytokines seem to be decreased in C-Vx treated group except for the Lix. Please check if there is any error in the presentation of the data. If I missed any information, please correct me.

2.      Figure 3 needs to include the statistical analysis of inflammatory cells in different samples, rather than only presenting representative images.

3.      If the authors want to demonstrate that the activation of NK cells is due to drug treatment, flow cytometry needs to be performed for examination.

4.      In my opinion, the relationship among the host, drug, and virus is not very clear in the manuscript. We all know that coronavirus infection in the host will inevitably lead to strong inflammation. Based on the results of animal experiments, the decrease in viral load in mice treated with drugs should lead to a reduction in the level of inflammation caused by viral infection. This logic is consistent with the data presented in figure 5. Just like the statement “we aimed to understand whether it helps immune cells to attenuate the functionality and cytokine storm by in vitro assays.” in line 100.

I am not sure whether the authors want to emphasize that C-Vx activates the host's immune system and increases cytokines to suppress the virus after intervention, or that C-Vx suppresses the virus, resulting in a natural reduction in related inflammatory factors. Therefore, in general, the manuscript should clearly state whether the changes in the level of inflammation are due to the drug's inhibition of the virus or the drug's direct effect on the host.

Author Response

We thank the reviewer for the important comments and tried to improve our manuscript as a response to the addressed questions.

 We have changed our manuscript title in the light of reviewer’s valuable comments and our new title of the manuscript is “Therapeutic effect of C-Vx substance in K18-hACE2 transgenic mice infected with SARS-CoV-2”

Our answers are as follows:

 Reviewer-1

Thanks to the authors for providing this manuscript. This research demonstrates the therapeutic potential of C-Vx against coronavirus infection in a mouse model. The phenotypic data from the in vivo experiment in this study provides a valuable reference for readers.

Major points:

  1. The authors mentioned “it was determined that the levels of various cytokines and chemokines were increased significantly in the C-Vx treated group compared to the control group in the 15th day (last day) (Figure 5B).” at line 176-178. However, the results shown in Figure 5 appear to be opposite. The detected cytokines seem to be decreased in C-Vx treated group except for the Lix. Please check if there is any error in the presentation of the data. If I missed any information, please correct me.

Thanks for reminding this important point, as you mention, cytokine levels increased in C-Vx group in day 15 as we were written in the results section. Unfortunately, this was incorrectly stated in the legend. This mistake was corrected in legend of Figure 5B.

  1. Figure 3 needs to include the statistical analysis of inflammatory cells in different samples, rather than only presenting representative images.

We are thankful for this important comment. The study employed a semi-quantitative evaluation. The condition of inflammation in the lungs was compared to that of the entire lung, and a number between 0 and 3 was assigned.

This is according to;

0: There is no inflammation.

1: Normal morphology, but slight erythrocyte and lymphocyte infiltration around the bronchioles.

2: The lung has moderate erythrocyte and lymphocyte infiltration.

3: Intense erythrocyte and lymphocyte infiltration and distorted morphology

The lung histopathology scoring of all animals was converted into statistical graph, new histogram was added to the Figure 2. Relative changes also were added to the text. Histopathology microscope images were presented as a supplementary Figure 1.

  1. If the authors want to demonstrate that the activation of NK cells is due to drug treatment, flow cytometry needs to be performed for examination.

Thanks for reminding this important point. We have changed the abstract and new version of abstract has been changed as “C-Vx is a bioprotective product designed to boost the immune system. This study aimed to determine the antiviral activity of the C-Vx substance against SARS-CoV-2 infection. The effect of C-Vx in K18-hACE2 transgenic mice against the SARS-CoV-2 virus was investigated. For this purpose, ten mice were separated into experimental and control groups. Animals were infected with SARS-CoV-2 prior to administration of the product to determine whether the product has a therapeutic effect similar to that demonstrated in previous human studies, at a histopathological and molecular level. C-Vx-treated mice survived the challenge, whereas the control mice became ill and/or died. The cytokine-chemokine panel with blood samples taken during the critical days of the disease revealed detailed immune responses. Our findings showed that C-Vx presented 90% protection against the SARS-CoV-2 virus infected mice. The challenge results and cytokine responses of K18-hACE2 transgenic mice matched previous scientific studies, demonstrating the C-Vx's antiviral efficiency.”

  1. In my opinion, the relationship among the host, drug, and virus is not very clear in the manuscript. We all know that coronavirus infection in the host will inevitably lead to strong inflammation. Based on the results of animal experiments, the decrease in viral load in mice treated with drugs should lead to a reduction in the level of inflammation caused by viral infection. This logic is consistent with the data presented in figure 5. Just like the statement “we aimed to understand whether it helps immune cells to attenuate the functionality and cytokine storm by in vitro assays.” in line 100.

We are thankful for this important comment C-Vx is a biological substance, however the mechanism is still not well known. It includes a SARS-CoV-2 gene fragment, an adjuvant, vitamins, and attenuated mumps and rubella viruses. We might speculate that CV-x may block the SARS-CoV-2 activity of the attenuated mumps and rubella viruses it carries, an effect that may be linked to the effect on mice. In general, immune cells and chemokines, cytokines etc. might play a protective role in inflammatory condition so, in the coming projects, we are planning to investigate the mechanism in detail.

Reviewer 2 Report

The article "Protective effect of C-Vx substance in K18-hACE2 transgenic 2 mice infected with SARS-CoV-2" explores the effectiveness of the C-Vx substance against SARS-CoV-2 infection. The applied methods seem sufficient to evaluate effectiveness in this early stage. The results are promising. However, there are several problems with the article:

- The introduction doesn't seem to fulfill its purpose. There is a generic part about COVID and immune reactions, but only a few lines about C-Vx. What is this compound? What does it do? Why do the authors think this compound could have beneficial effect? What is the hypothesis? These questions need to be answered in an introduction.

- The discussion is quite short and doesn't provide much explanation of the results

- The resolution of the figures are too low. Increase the DPI.

- On Fig 5 there is a mention of MWU test which is not in the statistical description, therefore it is not complete.

- There is nothing mentioned in statistics about how the data is checked for the assumptions of the applied tests.

- Lung pathology score looks like a categorical variable, t-test is not appropriate in that case

- If the researchers used fixed volume/mouse doses, than how did they account for the variability in mouse weight?

- How was the blood drawn from the mice? How much sample was used for the assays?

Overall, the quality of this paper does not reach the standards expected in a ~6IF journal.

- The article needs extensive English language editing

Author Response

We thank the reviewer for the important comments and tried to improve our manuscript as a response to the addressed questions. We have changed our manuscript title in the light of reviewer’s valuable comments and our new title of the manuscript is “Therapeutic effect of C-Vx substance in K18-hACE2 transgenic mice infected with SARS-CoV-2”

Our answers are as follows:

Reviewer-2

The article "Protective effect of C-Vx substance in K18-hACE2 transgenic 2 mice infected with SARS-CoV-2" explores the effectiveness of the C-Vx substance against SARS-CoV-2 infection. The applied methods seem sufficient to evaluate effectiveness in this early stage. The results are promising. However, there are several problems with the article:

- The introduction doesn't seem to fulfill its purpose. There is a generic part about COVID and immune reactions, but only a few lines about C-Vx. What is this compound? What does it do? Why do the authors think this compound could have beneficial effect? What is the hypothesis? These questions need to be answered in an introduction.

Thanks for reminding this important point. Patent (19) information and description of the C-Vx formula has been added to the relevant section as “According to some reports, innate immunity provided by some considered safe childhood vaccines, such as Bacillus Calmette-Guerin (BCG), Polio, Hemophilus influenza type-B, Measles-Mumps and Rubella (MMR), and pneumococcus, can provide significant protection against COVID-19 [12,26,27]. One of the reasons for the low prevalence of COVID-19 in children appears to be the high levels of immunity provided by childhood vaccinations [14]. However, the use of the relevant childhood vaccines, particularly in conjunction with the MMR vaccine, indicates that the MMR vaccine provides most of the protection against COVID-19 disease. There are other hypotheses that the SARS-CoV-2 surface protein shares 29% of its sequence with the rubella virus [27-28].”

- The discussion is quite short and doesn't provide much explanation of the results

Thanks for addressing this question. The relationship between our results and the MMR vaccine was emphasized in the discussion section.

- The resolution of the figures are too low. Increase the DPI.

We are completely re-designed all figures as advised by the reviewer, which we believe are more informative and representative. The resolution of all figures has been increased.

- On Fig 5 there is a mention of MWU test which is not in the statistical description, therefore it is not complete.

Thank you for this important comment. The statistic section has been edited according to your recommendation.

- There is nothing mentioned in statistics about how the data is checked for the assumptions of the applied tests.

Thank you for this important comment. The statistic section has been edited according to your recommendation.

- Lung pathology score looks like a categorical variable, t-test is not appropriate in that case

Thanks for your comment, however lung inflammation was scored from 0 to 3 (0: no inflammation, 1: low, 2: medium, 3: high) and in this way our data is categorical, so t-test was applied and the gross pathology results in the lung were significant (p=0.0141).

- If the researchers used fixed volume/mouse doses, then how did they account for the variability in mouse weight?

We thank for the reviewer for this important reminder. Since the newly developed C-Vx is a biological product, no volume per weight calculation was made. Animals weighing 20-22 g and living in the same environment were selected in each group and the same volume of C-Vx was used for all animals. This information has been added to the manuscript with new references supporting it.

- How was the blood drawn from the mice? How much sample was used for the assays?

Thanks for reminding this important point. Detailed explanation has been added to the relative session as “Tail blood was sampled before viral inoculation at the beginning of the study from experimental and control groups. An insulin injector was used to collect approximately 50 µl of blood from the lateral tail vein. 24 hours after the second viral inoculation, tail blood was drawn again. Since it is known that changes in cytokine and chemokine levels are observed at a significant level in 4-6 days of the disease, for control animals, this time interval is important [32]. Blood (0.5 ml) was taken from the animal while they were under anesthesia by inserting an insulin injector into the heart at a 20-30 degree angle just to the left of the xiphosternal midline. Mice were sacrificed by cervical dislocation following anesthesia with sevoflurane inhalation, and autopsies were conducted. Blood serum was obtained and stored at -20oC for cytokine measurements, which required 25 µl of serum per well.”

In accordance with your recommendation, we included the specifics of how we took the blood and the amount used to the relevant section.Overall, the quality of this paper does not reach the standards expected in a ~6IF journal.

We had assistance from a native English speaker and have tried to correct typos in the text, and we hope the new version of the article is suitable for the journal.

Round 2

Reviewer 1 Report

Thanks for the authors reply and improvement of the manuscript. In the research of natural and adaptive immunity in relation to the c-vx drugs, virus, and host, I look forward to the author conducting further in-depth research.

Reviewer 2 Report

I have no further comments.